# An evolutionary model identifies the main evolutionary biases for the evolution of genome-replication profiles

**Rossana Droghetti[1], Nicolas Agier[2], Gilles Fischer[2], Marco Gherardi[3], Marco Cosentino Lagomarsino[3,4]\***

[1]Dipartimento di Fisica, Università degli Studi di Milano, via Celoria 16, Milan, Italy; [2]Sorbonne Universitè, CNRS, Institut de Biologie Paris-Seine, Laboratory of Computational and Quantitative Biology, Paris, France; [3]Dipartimento di Fisica, Università degli Studi di Milano, via Celoria 16, Milan, Italy and INFN sezione di Milano, Milan, Italy; [4]IFOM Foundation, FIRC Institute for Molecular Oncology, via Adamello 16, Milan, Italy

**Abstract** Recent results comparing the temporal program of genome replication of yeast species belonging to the *Lachancea* clade support the scenario that the evolution of the replication timing program could be mainly driven by correlated acquisition and loss events of active replication origins. Using these results as a benchmark, we develop an evolutionary model defined as birth-death process for replication origins and use it to identify the evolutionary biases that shape the replication timing profiles. Comparing different evolutionary models with data, we find that replication origin birth and death events are mainly driven by two evolutionary pressures, the first imposes that events leading to higher double-stall probability of replication forks are penalized, while the second makes less efficient origins more prone to evolutionary loss. This analysis provides an empirically grounded predictive framework for quantitative evolutionary studies of the replication timing program.

**\*For correspondence:**
marco.cosentino-lagomarsino@
ifom.eu

**Competing interest:** The authors declare that no competing interests exist.

## Introduction

Eukaryotes, from yeast to mammals, rely on predefined 'replication origins' along the genome to initiate replication *Leonard and Méchali, 2013*; *Musiałek and Rybaczek, 2015*; *Ganier et al., 2019*; *Gilbert, 2001*, but we still ignore most of the evolutionary principles shaping the biological properties of these objects. Binding by initiation complexes defines origins as discrete chromosomal loci, which are characterized by multiple layers of genomic properties, including the necessary presence of autonomously replicating sequences, nucleosome depletion, and absence of transcription (*Méchali et al., 2013*; *Di Rienzi et al., 2012*). Initiation at origins is stochastic, so that different cells of the same population undergoing genome replication in S-phase will typically initiate replication from different origins *Bechhoefer and Rhind, 2012*; *Rhind et al., 2010*.

Initiation from a single origin can be described by intrinsic rates and/or licensing events *Hawkins et al., 2013*. Indeed, the genome-wide replication kinetics of a population of cells can be accessed experimentally by different techniques *Hawkins et al., 2013*; *Baker et al., 2012*; *Agier et al., 2013*. Recent techniques also allow to measure replication progression at the single-cell level *Müller et al., 2019*; *Hennion, 2020*. The estimation of key origin parameters from data requires minimal mathematical models describing stochastic origin initiation and fork progression *Retkute et al., 2012*; *Baker et al., 2012*; *de Moura et al., 2010*; *Zhang et al., 2017*. Typically, one can extract from the data origin positions, as well as estimated origin-intrinsic characteristic firing times or rates. Knowledge of origins

positions and rates makes it possible to estimate the 'efficiency' of an origin, that is, its probability of actively firing during S-phase, rather than being passively replicated.

Over evolution, a genome modifies its replication timing profile by 'reprogramming' origin positions and rates in order to maximize fitness under the constraints of the possible changes of these parameters that are physically and biologically accessible. Little is known about this process, and finding basic rules that drive origin evolution is our main focus here *Koonin, 2011*. The main recognized constraint determining negative selection is due to replication forks stalling between adjacent origins *Newman et al., 2013*; *Letessier et al., 2011*; *Cha and Kleckner, 2002*. If two converging replication forks stall with no origins in between them, it is generally agreed that replication cannot be rescued, and the event leads to cell death. Such deadly 'double stalls' can only happen with two converging forks generated from consecutive origins. A pioneering study by *Newman et al., 2013* used a combination of data analysis and mathematical models to understand the role of lethal double-stall events on origin placement. They found that the fork per-base stall probability affects the distance between neighbor origins, and the optimal distance distribution tends to a regular spacing, which is confirmed by experimental data. Thus, origin placement is far from a uniform random distribution (which would translate into an exponential distribution of neighbor origin distances). Instead, the regular lattice-like spacing that origin tend to take is reminiscent of particles repelling each other.

Due to the streamlined genome and the experimental accessibility, yeasts are interesting systems to study experimentally the evolution of replication programs. However, at the level of the *Saccharomyces* genus, the replication program is highly conserved *Müller and Nieduszynski, 2012*. Hence, until recently, no experimental account of the evolution of the replication program was available. Our collaboration has recently produced data of this kind *Agier et al., 2018* by comparing replication dynamics and origin usage of 10 distant *Lachancea* yeast species. This study highlights the dominance of origin birth-death events (rather than, e.g., chromosomal rearrangements) as the main evolutionary drive of the replication program changes and characterizes the main principles underlying origin birth-death events. Briefly, the fate of an origin strongly depends on its neighbourhood, particularly the distance from neighbor origins and their efficiency. Indeed, proximity to efficient origins correlates with weaker origin loss events. An evolutionary bias against weak origins could be due to the fact that their presence is neutral or even advantageous (e.g., in terms of reducing double stalls), but their advantage is not sufficiently high for them to survive drift. These findings open the question of capturing the relevant evolutionary biases acting on replication profiles in the framework of the empirical birth-death evolutionary dynamics, for which the data set *Agier et al., 2018* provides an empirical testing ground.

Here, we define a minimal evolutionary birth-death model for replication program evolution encompassing all the empirical observations made by *Agier et al., 2018*, and we use it to investigate the main evolutionary trade-offs that could explain the data.

## Results

### Experimental data motivate an evolutionary model for origins turnover

This section presents a reanalysis of the experimental data from *Agier et al., 2018*. We summarize the main results of that study and present additional considerations on the same data, which motivate the evolutionary model framework used in the following.

*Figure 1—figure supplement 1* recapitulates the *Lachancea* clade phylogenetic tree used in the analysis. The evolution of the temporal program of genome replication can be quantified by the divergence of the replication timing profiles across different species. Agier and coworkers found that timing profiles diverge gradually with increasing evolutionary divergence between species *Agier et al., 2018*. In principle, such divergence could be attributed to changes in the number, placement, and biological properties of all origins. However, a careful analysis of correlations (comparing the timing profiles and the activity of orthologous origins) shows that the main driver of program differentiation across species is the acquisition and loss of active replication origins. Specifically, the number of conserved origins decreases with increasing phylogenetic distance between species, following the same trend as the conservation of the timing profiles. This trend is the same in regions that are close to or away from breakpoints, pointing to a secondary role of genome rearrangements. In addition, the authors of *Agier et al., 2018* show that the differences in the mere number of origins and the median

difference in origin replication timing between pairs of species are nearly constant with phylogenetic distance, leading to exclude that origin reprogramming (rather than birth-death) plays a primary role in the evolution of the timing program.

Any model for the evolution of the replication program must (i) reproduce the empirical distribution of the inter-origin distances, (ii) reproduce the empirical distribution of the origin efficiencies, and (iii) account for the observed origin turnover dynamics. Previous analyses *Agier et al., 2018*; *Newman et al., 2013* have shown that origins are far from following a uniform distribution along the genome. *Figure 1A* shows that the inter-origin distance distribution robustly shows a unimodal shape across the 10 *Lachancea* species studied in *Agier et al., 2018*. Specifically, distributions for each species show a marked peak around 35 kbp. This peak corresponds to a typical inter-origin distance, which is strikingly invariant across all *Lachancea* species. *Figure 1B* shows the distribution of the efficiencies, which is defined as the probability to actively fire during the S phase, estimated for each origin in the *Lachancea* clade using *Equation 4* and a fit inferring the firing rates of all origins assuming a standard nucleation-growth model (see Materials and methods and *Zhang et al., 2017*). The single-species efficiency distributions show more variability across species than the inter-origin distance distributions, but they are consistent with a common shape and support.

As mentioned above, a key result of Agier and coworkers is the insight that the evolution of the replication program is mainly shaped by the birth-death process of replication origins. *Figure 1C–F* recapitulates the main quantitative results that characterize this process. Note that the analyses in *Figure 1C–F* have been performed on the six sister species of *Lachancea* clade since the other species pairs are too distant to perform a reliable identification of conserved, newly gained, and lost origins *Agier et al., 2018*.

*Figure 1C* shows a box plot of the distance from the nearest origin for all the conserved (dark red), newly gained (red), and lost (black) origins. Lost replication origins tend to be closer to their neighbors, much more so than newly gained or conserved origins. This observation reveals that the distance of an origin from its nearest neighbor is correlated to the loss rate of the same origin over evolution. This is an essential feature that any evolutionary model of this process must take into account *Newman et al., 2013*; *Agier et al., 2018*. More in detail, *Figure 1D* further quantifies the correlation between gain and loss events of neighboring origins by comparing the fraction of observed events of loss, gain, or conservation, given the state of the nearest origin (conserved, lost, or gained). The distribution of event types for origins that are nearest neighbors of a newly gained origin deviates significantly from the null expectation of random uncorrelated events (i.e., in a simple scenario where the fractions of conserved, newly gained, and lost origins are fixed to the empirical values, and birth and death events of neighboring origins are independent). The same non-null behavior is observed for origins that are nearest to a lost origin, with the roles of gain and loss events exchanged. In summary, successive birth/death or death/birth events happen more frequently in the same genomic location than expected by chance. Beyond such a spatial correlation along the chromosomal coordinate, the analysis illustrates that birth and death events are correlated in time as well (in fact, the analyzed evolutionary events took place in the terminal branches of the phylogenetic tree, and thus they must have been close in term of evolutionary time).

Finally, *Figure 1E and F* show that origins lying near loci where origins were recently lost are typically in the high-efficiency range of the distribution, and that lost origins tend to be less efficient than conserved origins. *Figure 1E* compares the distribution of the efficiency of lost, conserved, and newly gained origins with the distribution of efficiency of the nearest origins. The efficiency of origins neighboring a loss event is higher than average, while the efficiency of lost origins is lower than average. These results clearly support the influence of origin efficiency on origin death events. This is confirmed by *Figure 1F*, which shows the distribution of efficiency of all conserved and newly gained origins. For both classes, considering only those origins that are nearest neighbors to a recently lost origin yields an increase in the efficiency.

Different mechanisms could lead to the correlations described above. Overall, it is clear that origin strength is somehow 'coupled' to birth-death events. For example, conserved origins may become more efficient after the loss of neighbor origins, or the birth of new highly efficient origins could facilitate the loss of neighbors, or losing an origin could expedite the acquisition of a new origin nearby. Overall, these results reveal that the origin birth-death process is following some specific 'rules' that involve both inter-origin distances and origin efficiency.

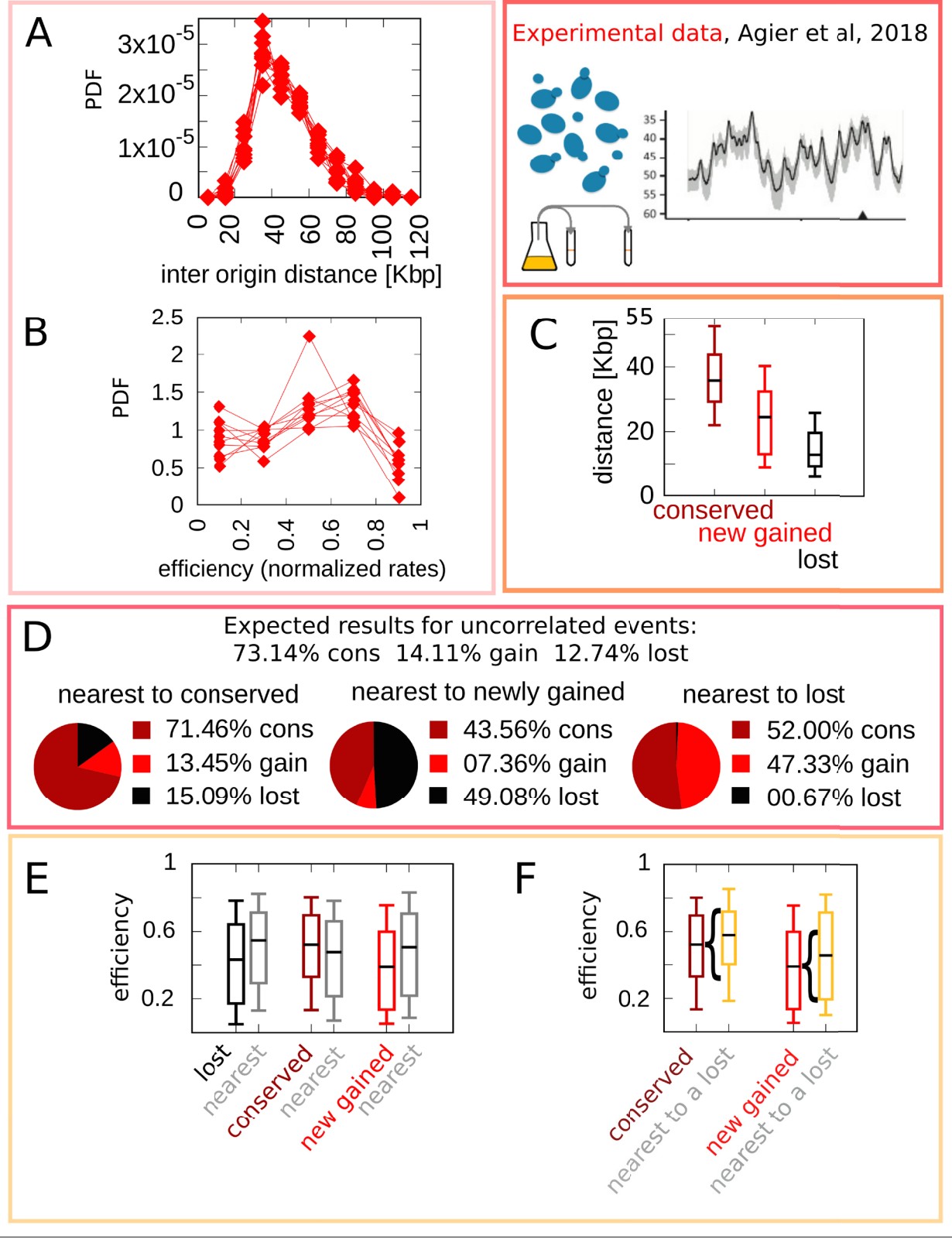

**Figure 1.** Experimental data motivate an evolutionary model for replication origins turnover. (**A**) Distribution of the distance between neighbor origins in 10 *Lachancea* species, each histogram refers to a different species (data from ***Agier et al., 2018***), and all the plots show a marked peak around 35 kbp. (**B**) Distribution of the efficiency (calculated from a fit, using ***Equation 4***) for all origins in 10 *Lachancea* yeast species ***Agier et al., 2018***. (**C**) From ***Agier et al., 2018***, box plot of the distribution of the distance from the nearest origin split by evolutionary events, for conserved (dark red), newly gained (red),

*Figure 1 continued on next page*

*Figure 1 continued*

and lost origins (black), estimated comparing six sister species of the *Lachancea* clade **Agier et al., 2018**. (**D**) Analysis of the origins that are nearest to conserved, newly gained, and lost, compared to the expected result if events were uncorrelated **Agier et al., 2018**. (**E**) Distribution of the efficiency of lost, conserved, and newly gained origins (respectively in black, dark red, and red) and their neighbors (gray). Note that the efficiency of lost origins is lower than average, while the efficiency of origins flanking a lost origin is higher. (**F**) Box plot of efficiency of all conserved and newly gained origins compared to those flanking a lost origin, which tend to be more efficient. Braces indicate subsampling (the box plots on the right side are defined by a subset of points of the box plots on the left). Box plots show the median (bar), 25–75 (box), and 10–90 (whiskers) percentiles. The data in panel (**C–F**) refers to the six sister species of the *Lachancea* tree.

The online version of this article includes the following figure supplement(s) for figure 1:

**Figure supplement 1.** The phylogenetic tree of the 10 *Lachancea* yeasts clade.

**Figure supplement 2.** The majority of new origins are born within a 20% distance from the midpoint of the associated interval.

**Figure supplement 3.** Experimental data on the evolutionary change of firing rates process.

**Figure supplement 4.** The decaying trend of the Spearman correlation coefficient defines a characteristic time for the firing rate resample.

Note that the results of *Figure 1E* might appear to be incompatible with *Figure 1D*, but they are not. *Figure 1E* shows that the efficiency of newly gained origins is lower than average, and *Figure 1D* shows that the majority of origins that are nearest to a locus with a recent loss event are newly gained. The apparent contradiction arises from *Figure 1E*, which shows that the average efficiency of origins close to a lost one is higher than average. This inconsistency is resolved by the analysis shown in *Figure 1F*, which shows that origins appearing close to recently lost ones are among the most efficient.

## A birth-death model including evolutionary bias from inter-origin replication fork double stalling recapitulates the main features of replication origin turnover

The joint stalling of two replication forks in the same inter-origin region along the genome is a well-characterized fatal event that may occur during S-phase. The frequency of this event in a clonal population clearly affects fitness. A previous modeling study **Newman et al., 2013** focusing on yeast demonstrated that, in order to minimize the probability of a double stall anywhere along the chromosome, origins must be placed in the most ordered spatial configuration, namely all the consecutive origins must be equidistant from each other. However, the previous study did not incorporate this principle into an evolutionary dynamics of origin turnover. Thus, the important question arises of whether the tendency to avoid double stalls is related to origin gain and loss. To address this question, we defined a birth-death model, rooted in the experimental observations discussed in the previous section. This 'double-stall-aversion model,' described in detail below, biases the turnover of replication origins in such a way that events (in particular, birth events) leading to a decreasing double-stall probability are promoted because they increase the fitness of the cell.

In the double-stall-aversion model, the extent to which the acquisition of a new origin changes the probability of a double stall $P_i^{\mathrm{DS}}$ depends on the length $l_i$ of the inter-origin region where the event occurred. This probability is therefore coordinate-dependent and can be derived by a procedure similar to the one carried out in **Newman et al., 2013** (see more details in Materials and methods),

$$P_i^{\mathrm{DS}} = 1 - (1 + \pi l_i) \exp(-\pi l_i) , \tag{1}$$

where $l_i$ is the length of the genome region between the (i+1)th and the ith origin and $\pi$ is the mean per-nucleotide fork-stall rate; we use the value from **Newman et al., 2013**, $\pi = 5 \times 10^{-8}$ per nucleotide. Note that the double-stall probability is completely independent from the origin firing rates and efficiency, and depends only on the distance between the origins.

In our simulations of the model (see Materials and methods for a more detailed explanation), the genome was represented as a vector of origins, identified by the position and the firing rate. The model is a discrete-time Markov chain, and for the double-stall-aversion variant the chain is specified by the following update rules:

- In each inter-origin region, the origin birth rate is biased by the value of the double-stall probability in that region. Specifically, the origin birth rate (per unit time) in the region , of length $l_i$ between the ith and (i+1)th origin is given by

$$b_i = N\bar{b}(P_i^{\mathrm{DS}})^\gamma l_i \, , \tag{2}$$

- where $P_i^{\mathrm{DS}}$ is the (constant) double-stall probability density in region  (**Equation 1**), $\bar{b}$ is the birth rate (per Mbp and per unit time) extracted from experimental data (see Materials and methods), and $\gamma$ is a positive real parameter that controls the strength of the bias. $N$ is a normalization factor added to match the empirical birth rate $\bar{b}$. Newborn origins are placed in the middle of the inter-origin region .
- Death (i.e., loss of origins) is unbiased and occurs at random origins with rate $\bar{d}$ (estimated from experimental data, see Materials and methods), regardless of their efficiency or their neighbor's efficiency.

The justification for the assumption that newborn origins are placed at midpoints in the model ultimately comes from data (**Figure 1—figure supplement 2**) where a strong bias in this direction is found. Relaxing this assumptions has consequences on the distance distribution and leads to poorer-performing models. We interpret this bias as the result of a faster (hence undetectable in our data) evolutionary process that counter-selects origins far from midpoints.

Firing rates in the model evolve by reshuffling of the empirical firing rate distribution, with a time scale that is set empirically (see Materials and methods and **Figure 1—figure supplements 3 and 4**). On shorter time scales, firing rate changes are likely more gradual, making firing rate evolution similar to a diffusion process. However, such changes are not quantifiable in our data set, which would leave the model with many extra parameters (a firing rate diffusion constant and bounds to set the empirical distributions) that are very difficult to estimate. Additionally, the firing rate distributions of the conserved (thus older) origins and of newborn (younger) ones are quite similar (**Figure 1—figure supplement 3**), and this condition is not generally met under a simple diffusive process.

**Figure 2** shows the simulation results of the model with best-fitting parameter values (see Materials and methods and **Figure 2** for other parameter values). **Figure 2A and B** show that the double-stall-aversion model reproduces the two main 'structural' features of yeast genome, namely the inter-origin distance distribution and the origin efficiency distribution. Additionally, **Figure 2C and D** show that the same model reproduces the observed correlations between the inter-origin distance and origin birth-death events, as well as the correlation between birth-death events and nature of the neighbor origins observed in the data (conserved, newly gained, or lost).

## The double-stall hypothesis alone fails to capture correlations of origin turnover with efficiency

In spite of the good performance of the double-stall-aversion model in explaining the empirical marginal distributions, we find that it fails to reproduce the observed correlations between the efficiency of an origin and the recent history of the nearest ones. **Figure 2E** shows very faint variations in efficiency of origins that are nearest neighbors to origins of different evolutionary fate. In particular, the observed huge divergence in efficiency between lost origins and their neighbors is absent in the model simulations. Note that **Figure 2** and Figure 4 show that in the double-stall-aversion model origins nearest to a loss event are slightly more efficient than average. This trend is due to the fact that after an origin is lost, its neighbors are subject to lower interference and automatically become more efficient. However, Figure 4 shows that this null trend is too weak to explain the experimental data. These considerations indicate that a model without a direct mechanism linking the efficiency of an origin to the birth-death events of its neighbors cannot reproduce the data.

## Double-stall aversion and interference between proximate origins explain the correlated evolution of origin presence and efficiency

Based on the above considerations, we defined a joint model that takes into account both the evolutionary pressure given by the double-stall probability and the direct effect of origin efficiency on birth-death events.

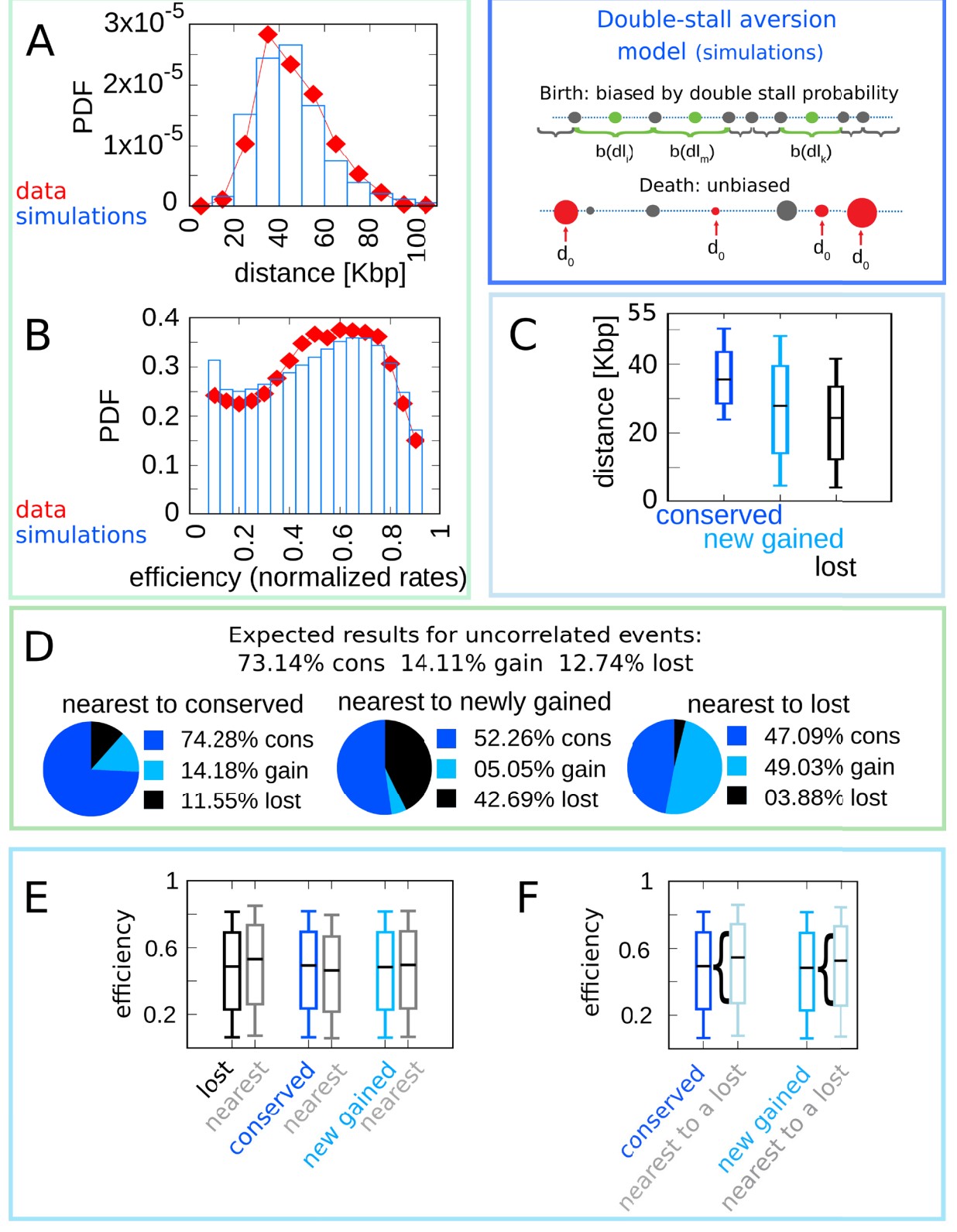

**Figure 2.** The double-stall-aversion model reproduces origin turnover and distributions but fails to capture correlations between origin turnover and origin strength. The plots show the simulations of the best-fitting double-stall-aversion model compared with empirical data. (**A**) Inter-origin distance distribution in simulated species (blue bars) compared to the empirical distribution for the 10 *Lachancea* species (red diamonds). (**B**) Origin efficiency distribution in simulated (blue bars) vs. empirical species (red diamonds). (**C**) Box plot of the distance from the nearest origin split by evolutionary events,

*Figure 2 continued on next page*

*Figure 2 continued*

that is, for conserved (dark blue), newly gained (blue), and lost origins (black) for simulated species. (**D**) Fraction of origins that are nearest to conserved, newly gained, and lost for simulated species compared to the expected result for uncorrelated events. (**E**) Box plot of efficiency of lost, conserved, and newly gained origins (respectively in black, dark blue, and blue) and their neighbors (gray) in simulated species. The six distributions show very little variation. (**F**) The efficiency of all conserved and newly gained origins compared to the ones flanking a lost origin. Braces indicate subsampling. Box plots show the median (bar), 25–75 (box), and 10–90 (whiskers) percentiles. Simulation parameters (see Materials and methods): $\gamma = 2.4$ overall birth and death rate $\bar{b} = 13.6 Mbp^{-1} t^{-1}$, $\bar{d} = 0.61 t^{-1}$ and firing rate resampling rate $R = 0.92 t^{-1}$, where $t$ is measured by protein-sequence divergence. Panels (**A**) and (**B**) were generated using data from approximately 320,000 simulated origins, while panels (**C–F**) were built using data from about 60,000 birth and death events and 240,000 conservation events.

Specifically, this model is defined as follows:

- The birth process is the same as in the double-stall-aversion model described above: the birth rate is biased by the double-stall probability in each inter-origin region *Equation 1*, and newborn origin are placed in the middle of the region.
- Death of an origin is biased by its efficiency: less efficient origins are more easily lost. Specifically, the death rate (per unit time) for the ith origin is

$$d_i = N\bar{d}\exp(-\beta \, \mathrm{eff}_i), \tag{3}$$

- where $eff_i$ is the efficiency of the ith origin, *Equation 4*, $\bar{d}$ is the mean death rate extracted from experimental data (see Materials and methods). The positive parameter β tunes the interaction strength: the larger β, the steeper the dependence of $d_i$ on $eff_i$. The normalizing factor $N$ is chosen so as to match the empirical total death rate.

We note that the bias parameters β and γ are not inferred based on branch data, but on distributions of extant species (see Materials and methods).

*Figure 3* gathers plots of the structural features (distribution of inter-origin distances and efficiencies, *Figure 3A–B*) and the evolutionary correlations involving efficiency, evolutionary fate, distance to nearest neighbor, and fate of nearest neighbor (*Figure 3C–F*). Overall, the joint model reproduces all the observations considered here regarding the layout of origins and their evolutionary dynamics, indicating that the experimental data can be rationalized by a fitness function that includes both the detrimental effects of nonreplicated regions and the evolutionary cost of maintaining inefficient replication origins.

In particular, the coupling between the efficiency of an origin and the death rate of its neighbors, through the probability of passive replication, reproduces the empirical correlations shown in *Figure 1*. *Figure 4* summarizes this crucial point of comparison between the joint efficiency/double-stall-aversion model and the pure double-stall-aversion case. The three plots compare efficiency distributions of lost, conserved, and newly gained origins (red for the data, blue for the models) with those of their neighbors (gray). Comparison of these plots shows that only the joint model reproduces the differences in efficiency of lost origins and their neighbors.

In order to show that the stall-aversion and interference model has better quantitative agreement with the data, we also performed a simplified likelihood ratio analysis. The full likelihood of the model is complex, but we have defined 'partial' likelihoods for the joint and the double-stall-aversion model just taking into account the marginal probabilities shown as box plots in *Figure 4*, *Figure 4—figure supplement 1* (see Materials and methods). *Figure 1* shows that the joint model performs better for all the four chosen features. In our view, the qualitative difference shown in *Figure 4* may be taken as a stronger argument in favor of the combined model in the sense that, beyond any quantitative agreement relying on parameters, the additional ingredient of a coupling between origin birth-death dynamics and origin rates is needed to explain the data.

## The joint efficiency/double-stall-aversion model correctly predicts origin family divergence

Having established that the joint model is required to reproduce observations on single lineages, we turned to its predictions on observations that require knowledge of the whole phylogenetic tree, such as origin evolutionary families, defined as sets of orthologous origins *Agier et al., 2018*.

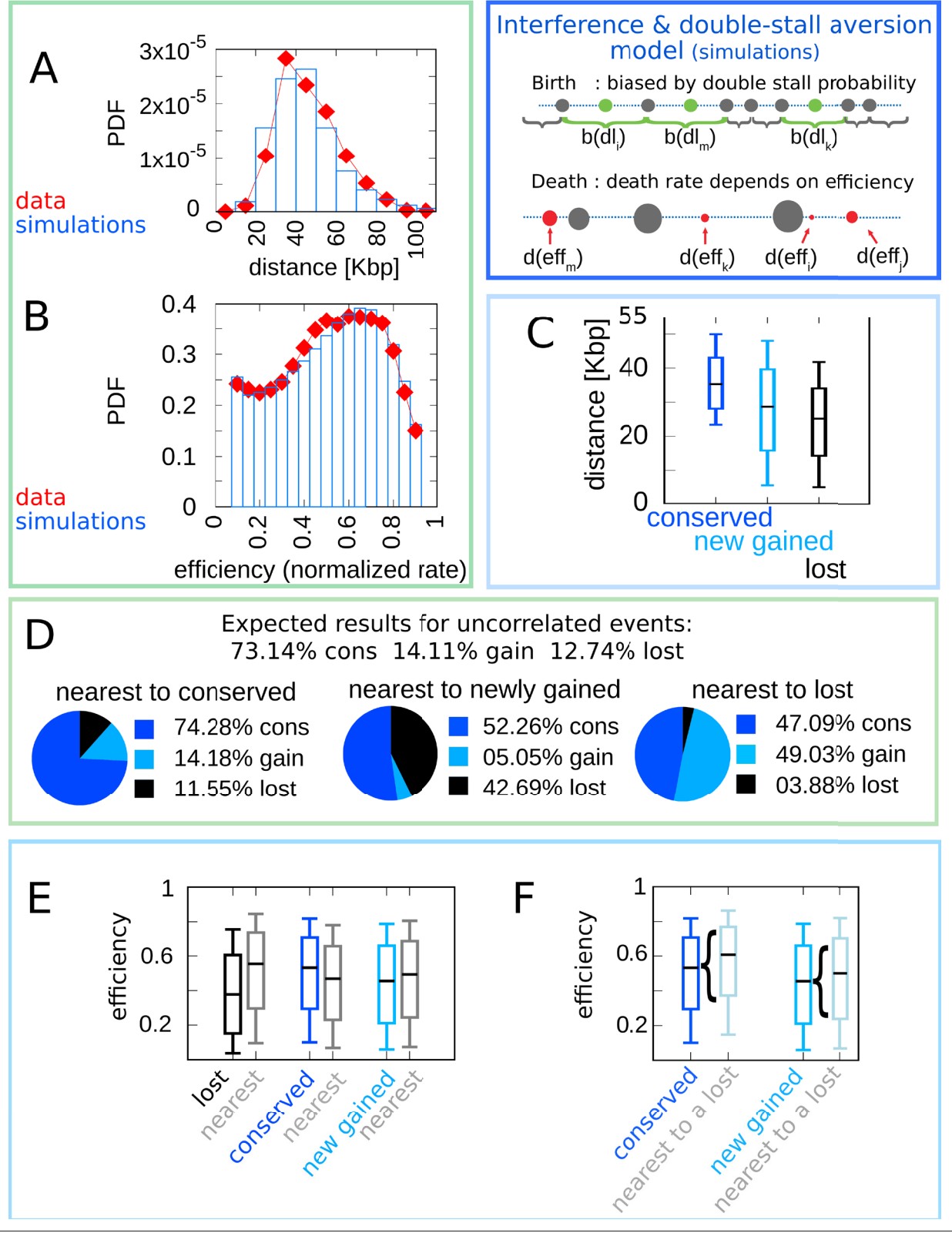

**Figure 3.** A model where both fork stalling and interference affect fitness explain the correlations between origins of evolutionary events. Result of the joint model best-fitting simulation compared with empirical data. (**A**) Inter-origin distance distribution in simulated species (blue bars) vs. empirical distribution for the 10 *Lachancea* species (red diamonds). (**B**) Origin efficiency distribution in simulated (blue bars) vs. empirical species (red diamonds). The agreement between simulation and experimental data shows that this joint evolutionary model reproduces the typical structural features of a yeast

*Figure 3 continued on next page*

*Figure 3 continued*

genome. (**C**) Box plot of the distance from the nearest origin split by evolutionary events, that is, for conserved (dark blue), newly gained (blue), and lost origins (black) for simulated species. (**D**) Fraction of origins that are nearest to conserved, newly gained, and lost for simulated species compared to the expected result for uncorrelated events. (**E**) Box plot of efficiency of lost, conserved, and newly gained origins (respectively in black, dark blue, and blue) and their neighbors (gray) in simulated species. (**F**) The efficiency of all conserved and newly gained origins compared to the ones flanking a lost origin. Braces indicate subsampling. Box plots show the median (bar), 25–75 (box), and 10–90 (whiskers) percentiles. Panels (**D**–**F**) show that the model correctly reproduces the correlation between origin birth-death events over evolution and efficiency of the nearest origin. Simulation parameters (see Materials and methods): $\gamma = 2.2$, $\beta = 1.9$, overall birth and death rate $\bar{b} = 13.6 Mbp^{-1}t^{-1}$, $\bar{d} = 0.61t^{-1}$, and rate of origin firing rate reshuffling $R = 0.92t^{-1}$, where $t$ is measured by protein-sequence divergence. Panels (**A**) and (**B**) show data from approximately 600,000 simulated origins, while panels (**C**–**F**) data from about 100,000 birth and death events and 500,000 conservation events.

The online version of this article includes the following figure supplement(s) for figure 3:

**Figure supplement 1.** Linear chromosomes do not alter significantly the model outcomes.

We thus set up a simulation of the model on a cladogenetic structure, fixed by the observed structure of the *Lachancea* clade phylogenetic tree (see Materials and methods for the simulations details). The outputs of each run in such simulations are nine different simulated genomes whose lineages are interconnected in the same way as the empirical species, and each branch follows the empirical divergence. We stress that these simulations just include intersecting lineages whose branched structure corresponds precisely to the lineages of the empirical tree. The phylogenetic structure does not emerge from the simulation as our model does not describe speciation. The model for the tree can simulate nine species, all the species except for *Lachancea kluyveri*, as this species was used as outgroup for the computation of the length of the tree branches *Agier et al., 2018*. We have repeated all the analyses on these simulations and verified that all the previous results hold (*Figure 5—figure supplement 1*). We then turned to other independent predictions of the joint model, which could be compared to measurements in *Agier et al., 2018*.

*Figure 5A* reports the dynamics of origin families. As reported in *Agier et al., 2018*, origins that belong to larger evolutionary families tend to have a higher efficiency compared to origins in smaller families, which is possibly due to the fact that, on average, high-efficiency origins tend to survive longer. Note, however, that there is no deterministic relation between family size and origin age because the relationship between these two is determined by the structure of the phylogenetic tree. Indeed, two families of the same size may have roots in different points of the tree, and thus the origins belonging to them may have very different ages. Thus, the prediction of the relation between origin efficiency and origin family size is not trivial. *Figure 5A* shows the results for the origin efficiency for families of varying size, comparing the experimental data and 100 different runs of the simulation.

As a second step, we have considered the model prediction for the divergence of the shared origins in two species descending from a common ancestor. Specifically, we asked whether the number of origin death events occurring in two branches of the tree could justify the number of common origins in the two species. Indeed, whenever in a pair of species the number of shared origins is lower than the number of origins belonging to their common ancestor, this discrepancy must be due to the evolutionary loss events. These events are predicted by our model to be correlated in diverging species due to the common ancestry and the coupling of loss events to origin efficiency and distance. This correlation should lower the number of shared origins losses compared to a null expectation where loss events are not correlated. *Figure 5B* shows that the model correctly predicts the divergence in the number of shared origins lost during evolution without any parameter adjustment. We also verified that, as expected, a null evolutionary model is not able to reproduce this feature. The null model fixes in each branch of the simulated tree the same number of birth and death events that are present in the corresponding branch of *Lachancea* tree, but these events occur uniformly along the genome. The difference between the null model and the evolutionary model predictions shown in *Figure 5B* is a consequence of correlated origins losses due to the common genome structure, in terms of origins positions and efficiencies, that each pair of species inherit from their common ancestor.

We note that birth and death rate are inferred as global parameters, ignoring correlations. Despite this, *Figure 5B* shows that the model reproduces the higher correlation in birth and death events in closer-related branches than in distant branches as a consequence of the common positions and firing rates of the origins in the ancestor.

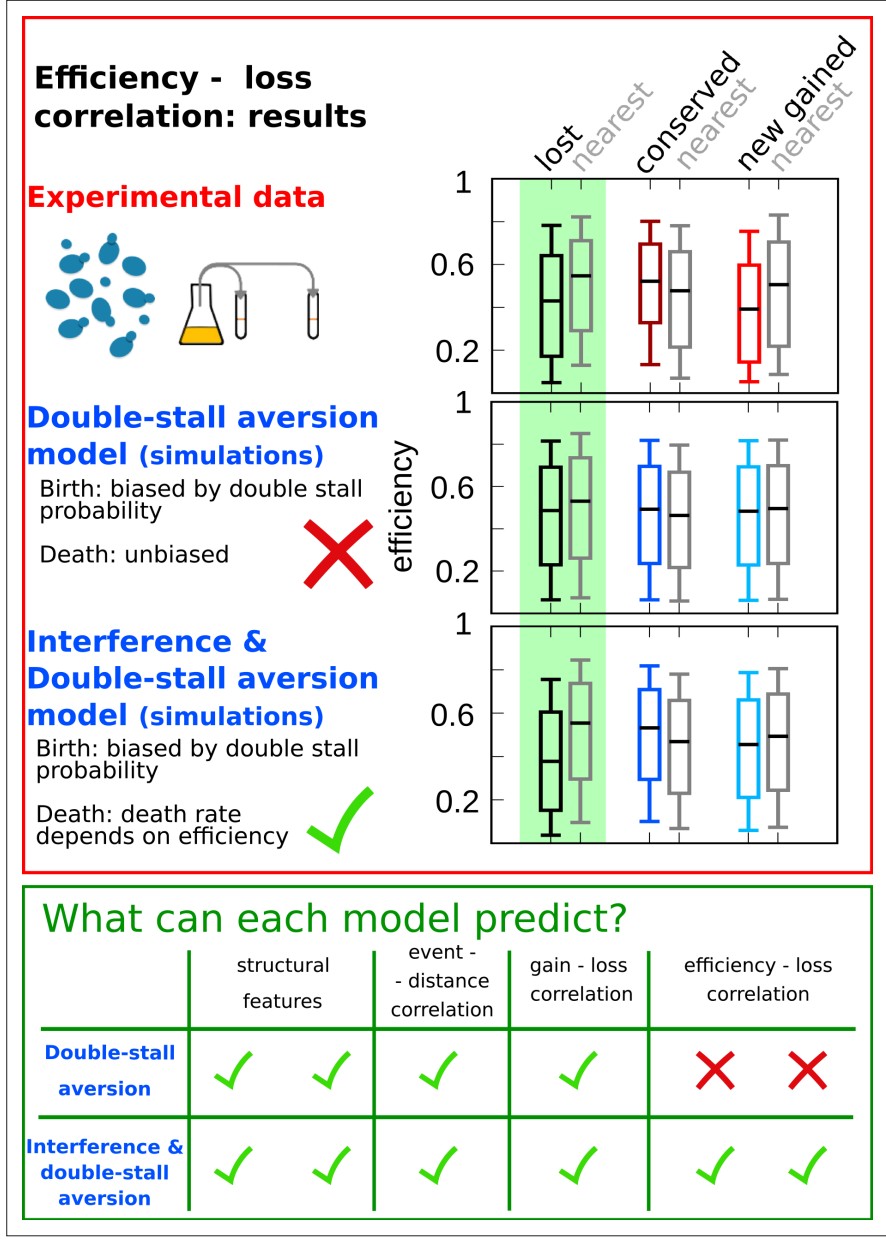

**Figure 4.** Comparison of model predictions for the correlations of origin birth-death events. The plots in the red upper box compare efficiency distributions of the best-fitting simulation of the two different models (bottom and central panels) with experimental data (top panel). Comparison of the box plot of efficiency of lost, conserved, and newly gained origins (red for the data, blue for the models) shows better agreement of the joint efficiency/ double-stall-aversion model (bottom panel) with the experimental data. Hence, the joint model reproduces well the correlation between evolutionary birth-death events of origins and efficiency of the nearest origin, while the double-stall-aversion model fails. Box plots show the median (bar), 25–75 (box), and 10–90 (whiskers) percentiles. Simulation parameters for the joint model (see Materials and methods): $\gamma = 2.2$, $\beta = 1.9$, and for the double-stall-aversion one: $\gamma = 2.4$. General parameters: overall birth and death rate $\bar{b} = 13.6 Mbp^{-1}t^{-1}$, $\bar{d} = 0.61t^{-1}$ and rate of origin firing rate reshuffling $R = 0.92t^{-1}$, where $t$ is measured by protein-sequence divergence. In the green lower box, we compare the predictive power of the two models for each of the tested feature of the experimental data. The box highlights that both the double-stall aversion model and the joint efficiency–double-stall model are able to reproduce the structural features of the genome. Also, the correlation between events–distance from the nearest and event–event of the nearest are correctly predicted by both models. The important difference between the two proposed models is found for the correlation between evolutionary events and origin efficiency, which is predicted and can be explained solely by the joint model.

*Figure 4 continued on next page*

*Figure 4 continued*

The online version of this article includes the following figure supplement(s) for figure 4:

**Figure supplement 1.** The efficiency mechanism is necessary to reproduce the correlation between firing rates and evolutionary events.

**Figure supplement 2.** Analytical predictions for the inter-origins distance distribution falsify the scenario whereby interference alone drives replication program evolution.

## Discussion

Overall, this study provides a framework to study replication program evolution driven by replication origin birth-death events and demonstrates that both fork stalling and efficiency shape the adaptive evolution of replication programs. The model framework is predictive and falsifiable, and it can be used to formulate predictions on the phylogenetic tree. In future studies, it would be interesting to explore the predictions for the evolutionary dynamics under perturbations, such as evolution under increased replication stress or conditions where fork stalling becomes more frequent. Additionally, the framework can be used to discover specific trends, such as different evolutionary dynamics of specific genomic regions (subtelomeres *Yue, 2017*, regions containing repeats, etc. *Arbona et al., 2018*;

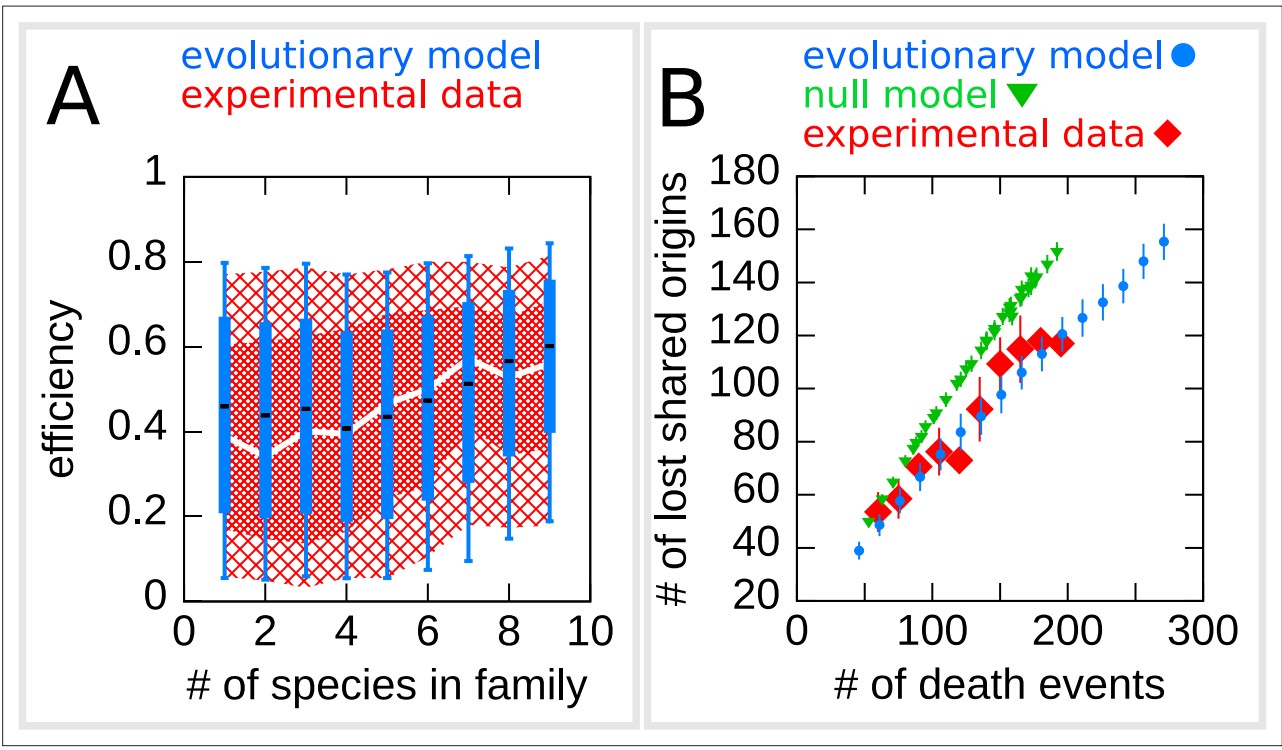

**Figure 5.** The efficiency/double-stall-aversion model predicts origin divergence. The plots compare predictions of the evolutionary model on the extent of origin divergence (simulations of the *Lachancea* phylogenetic tree) with empirical data. (**A**) Box plot of origins efficiency distributions split by family size. The plot compares origin families (sets of orthologous origins) in the nine *Lachancea* species (white line and red shaded areas) and simulated species (blue boxes, for 100 simulation runs). Medians are shown as white line for data, black bar for simulation, 25–75 percentiles as shaded area for data, box for simulation, and 10–90 percentiles as coarse shaded area for data, whiskers for simulation. (**B**) Origin divergence measured by the number of origins in the common ancestor that were lost in a pair of species, plotted as a function of total origin loss events. The plot compares model simulations (blue circles, 100 simulation runs), the experimental data (red squares), and a null model that shuffles the empirical birth-death events in each branch (green triangles, 1000 simulation runs). Error bars are standard deviations on y-axis values. Simulation parameters (for the evolutionary model, see Materials and methods): $\gamma = 2.2$, $\beta = 1.9$, overall birth and death rate $\bar{b} = 13.6 Mbp^{-1}t^{-1}$, $\bar{d} = 0.61t^{-1}$, and rate of origin firing rate reshuffling $R = 0.92t^{-1}$, where $t$ is measured by protein-sequence divergence.

The online version of this article includes the following figure supplement(s) for figure 5:

**Figure supplement 1.** The joint efficiency/double-stall-aversion model simulated on a cladogenetic structure reproduces all the results found for a single lineage.

**Figure supplement 2.** Simulations and empirical data show a similar variability in number of death and birth events across branches of the tree.

*Boos and Ferreira, 2019*), role of genome spatial organization *Marchal et al., 2019*, and correlated firing of nearby origins.

A general question concerns the predictive value of the model proposed here on out-of-sample data. *Figure 5* shows that fit-independent predictions apply across the tree. Importantly, the model is based on simple global parameters and not fine-tuned on local features of the tree. To underline this point, we verified that a model fit using only the subtree between LADA a LAWA yielded similar parameters. Clearly, we cannot exclude that the values of the birth and death rate, and also the bias parameters γ and β, could be *Lachancea*-specific, while we speculate that the conclusions on the relevant evolutionary mechanisms might apply more generally.

The previous approach by *Newman et al., 2013* described the evolution of origin distance as an optimization process that minimizes double fork-stall events, without attempting to characterize explicitly the evolutionary dynamics. Such approaches are limited compared to the framework presented here because they can predict only the origin distance distribution, and they do not allow any prediction regarding origin and replication program evolution along lineages and across phylogenetic trees. In accordance with the results of Newman et al., we confirm that double-stall events are a primary driver of the evolution of replication programs, and we frame this finding into the empirically measured birth-death evolutionary dynamics of replication origins. Additionally, we show that next to fork-stall events, origin efficiency plays an important role in shaping the evolutionary landscape seen by a replication timing profile.

What could be the mechanisms coupling efficiency to origin birth-death? The actual process of origin death could be nearly neutral *Koonin, 2016* as low-efficiency origins are – by definition – rarely used, and unused origins over evolutionary times are more prone to decay in sequence, and consequently in firing rate until they disappear. Equally, a newborn origin close to a very strong one (which would make the newborn origin relatively inefficient) could be used rarely. This would make this origin relatively less likely to establish over evolutionary times compared to an isolated newborn origin. However, rarely used origins could be essential in situations of stress (and in particular they could resolve double-stall events). Finally, a fitness cost for maintaining too many origins might set up an overall negative selection preventing a global increase in origin number *Zhang et al., 2017*; *Karschau et al., 2012*; *Das et al., 2015*.

## Materials and methods
### Data

The experimental data used in this work come from *Agier et al., 2018*. In particular, we made use of the data regarding the replication origins. For each origin in each of the 10 *Lachancea* species, this data set includes the chromosome coordinate and firing rate, and the inferred birth and death events occurred in the branches of the phylogenetic tree shown in *Figure 1—figure supplement 1*. Focusing on the terminal branches of the tree and on the extant replication origins, this study defines three categories of origins: (i) 'conserved' origins (which survived from the last ancestor), (ii) 'newly gained' origins gained in the last branch of the phylogenetic tree, and (iii) 'lost' origins, which were present in the last ancestor species and are not present in the terminal branch. Properties of the lost origins (e.g., position and firing rate) are inferred from the projection of the corresponding ones on the closest species, keeping into account synteny. Since the synteny map is less precise in distant species, the information on the origins events is only available for the six sister species in the tree, which belong to the three closest species pairs, highlighted with the red shaded area in *Figure 1—figure supplement 1*.

### Computation of the efficiency

Origin efficiency was defined as the probability of actively firing during S phase (or, equivalently, the probability of not being passively replicated by forks coming from nearby origins). In practice, we computed it by the following formula:

$$\text{eff}_i = (1 - P_{i,i-1})(1 - P_{i,i+1}) \,, \tag{4}$$

where $P_{i,i+1}$ and $P_{i,i-1}$ are the probabilities for the ith origin of passive replication respectively from the (i+1)th and (i-1)th origins. Note that this efficiency formula *Equation 4* is an approximation that

only takes into account the possibility to be passively replicated by neighbor origins, neglecting the influence of other nearby origins. Following *Agier et al., 2018*, for computing the efficiency we assumed that the origin firing process has constant rate *Zhang et al., 2017*, and we thus obtain the following closed expressions for the probabilities of passive replication:

$$P_{i,i+1} = \frac{\lambda'_{i+1}}{\lambda'_{i+1}+\lambda'_i} \exp\left[-\lambda'_i \frac{|x_{i+1}-x_i|}{v}\right] , \tag{5}$$

and

$$P_{i,i-1} = \frac{\lambda'_{i-1}}{\lambda'_{i-1}+\lambda'_i} \exp\left[-\lambda'_i \frac{|x_{i-1}-x_i|}{v}\right] . \tag{6}$$

In the above equations, $v$ is the typical velocity of replication forks, $x_i$ is the ith origin chromosome coordinate, and $\lambda'_i$ is the ith origin firing rate divided by the mean firing rate of the species the origin belong to. The raw firing rates in the data are affected by the different physiology of the nine *Lachancea* species in the experimental growth conditions (which were the same for all the species). In order to reduce these differences, we normalized the rates by their average for each given species. For this reason, we did not make use of the origin efficiency data already present in *Agier et al., 2018*.

## Computation of the double-stall probability

The probability $P_i^{DS}$ that two converging forks stall is easily computed in the limit where the stall probability per basepair is small and the number of basepairs is large. Under these assumptions, stalling is a Poisson process with rate (per basepair) π. $P_i^{DS}$ can be written in terms of the probability $P^S(x)$ that a single fork stalls after replicating $x$ nucleotides:

$$P_i^{DS} = \int_0^{l_i} dx \int_0^{l_i-x} dy \, P^S(x)P^S(y) , \tag{7}$$

where $l_i$ is the length (number of basepairs) of the ith inter-origin region. Imagine two converging replication forks starting from origins and $i + 1$: the two integration variables $x$ and $y$ represent the number of basepairs that each fork replicates before stalling. By using the Poisson process result $P^S(x) = \pi \exp(-\pi x)$ and performing the integration, one obtains the result in *Equation 1*.

## Evolutionary model

We defined origin birth-death models incorporating different evolutionary biases. In these models, the genome is described as a one-dimensional circle with discrete origin location $x_i$, where the length of the genome is equal to the average genome length in *Lachancea* clade ($10.7Mbp$). We made use of a circular genome in order to avoid border effects. In the model, the set of origins change over evolution by three basic (stochastic) processes, birth of an origin in a certain genome region, origin death, and change of origins firing rate. We have verified that choosing linear chromosome does not alter significantly our findings, although it affects the distances between origins close to chromosome ends (*Figure 3—figure supplement 1*).

Overall origin birth/death rates were estimated from the data as follows. To estimate the overall birth rate $\bar{b}$, we considered, for all the terminal branches of the phylogenetic tree, the number of birth events $N_b$, the genome length of the corresponding species $L$, and the length of the tree branch $T$, and divided $N_b$ by $LT$. Then we averaged over all terminal branches. To estimate the overall death rate $\bar{d}$, we followed a similar approach, taking the number of death events $N_d$ in the terminal branches, the length of the branch $T$, and the number of origins in the corresponding species $n_{ori}$, then computing $N_d T^{-1} n_{ori}^{-1}$ for all the terminal branches and averaging these values. The final results for overall birth and death rates from the origin birth-death events across the *Lachancea* clade are $\bar{b} = 13.5627Mbp^{-1}t^{-1}$ and $\bar{d} = 0.612287t^{-1}$.

We verified that the assumption of constant rates was consistent with the empirical variability of the numbers of birth and death events per unit time along different branches of the tree by comparing simulations with data. *Figure 5—figure supplement 2* shows that simulations and empirical data present similar spreading.

The process by which origin firing rates change over evolution was described as stochastic, with every origin having a fixed probability per unit time of changing its firing rate, given by $R = 0.92t^{-1}$,

a value fixed from experimental data (see Appendix 1 and *Figure 1—figure supplement 4*). When a firing rate changes, it is resampled from the distribution of all the empirical normalized firing rates computed using the data in *Agier et al., 2018* (see Appendix 1 and *Figure 1—figure supplement 3* for more details).

## Simulations
### Code availability
The code used to run the simulations, together with instructions to run it, was shared as a repository on Mendeley data (*Droghetti, 2020*).

### Algorithm
The prediction of the different evolutionary models was derived numerically, making use of custom simulations written in C++, which implement the origin birth-death dynamics as a Gillespie algorithm (*Gillespie, 1976*). Every model variant was required to reproduce the experimental overall rates, $\bar{b} = 13.5627 Mbp^{-1}t^{-1}$ for origin birth, $\bar{d} = 0.612287t^{-1}$ for origin death, and $R = 0.92t^{-1}$ for firing rate change. We simulated the three processes defining the model as follows. (i) the birth process has a common definition for the stall-aversion and joint model. The algorithm first tests each subsequent inter-origin region, calculates the birth probability from *Equation 2*, and stores the results. Subsequently, it computes the normalization factor $N$ in order to match the empirical birth rate per nucleotide $\bar{b}$. Finally, it samples all the inter-origin regions drawing birth events from the computed birth probability (*Equation 2*). New origins are placed the midpoints of the tested intervals. (ii) The death process is different for the stall-aversion model (unbiased) and the joint model (related to the origin efficiency). In the joint model, the algorithm first calculates the death rate for each origin using *Equation 3* and stores the results. Subsequently, it computes the normalization factor $N$ in order to match the empirical mean death rate $\bar{d}$. Finally, it samples all origin drawing death events from the computed death probability. For the unbiased process (stall-aversion model), the dynamics is identical, but all the origins have the same death rate $\bar{d}$ so that the algorithm can skip the calculation of $N$. (iii) The process updating origin firing rates over evolutionary times is common to all model variants. The probability of update per origin per unit time is $R$. Origins are sampled for each time step and assigned a new rate uniformly extracted from the empirical distribution of all normalized firing rates with probability $Rdt$.

During the simulation, the genome configuration (chromosome position, firing rate, efficiency for each origin) is known at each time step, which matches the empirical time (tree branch length, measured by protein-sequence divergence). For simulating single lineages, we started with a collection of 50 origins, with positions and firing rate uniformly drawn from all the possible ones. Rapidly, the inter-origin distances distribution, the efficiency one, and the number of origins reach a steady state (for the number of origins, set by the balance of birth and death rate, and characterized by approximately 225 origins). Configurations, including birth-death events, were printed at regular time intervals after steady state is reached. The time interval between prints is chosen to be equal to the average length of the *Lachancea* phylogenetic tree terminal branches in order to compare single-lineage simulations with empirical data. For simulations on a phylogenetic tree, after one species reaches the steady state, it is used as a root. To reproduce the empirical branching structure of the tree, we run the simulation, one for each branch of the phylogenetic tree, each time starting from the species at the previous branching point, for a period that matches the length of the branch. If the simulated branch is terminal, then the configuration corresponds to one of the empirical species, otherwise it corresponds to a 'branching-point species' and it can be used as a starting point for other simulations. Each simulation run gives nine different simulated species with the same cladogenetic structure as the empirical species (*Figure 1—figure supplement 1*).

### Fitting procedure
The biased birth-death processes in the simulations rely on some parameters to tune the strength of the bias, which are the only parameters to fix by a fit, since all the other parameter values are fixed empirically. In the joint model, there are two free parameters, γ and β, that tune respectively the strength of the bias on the origin birth and on the origin death process. For a discrete set of parameter pairs spanning realistic intervals, we run hundred different simulations, each starting with a randomized genome. Considering the simulated species for all the pairs of parameter values, we quantify

the discrepancy with experimental data by evaluating the L1 distance of the normalized histogram of efficiency and inter-origin distances. This quantity is a number between 0 and 2, 0 if the histograms perfectly overlap and 2 if they have completely different supports. For each pair of parameters, the analysis gives two values of discrepancy. We choose the value of γ (the parameter that tunes the bias on the birth rate based on double-stall aversion) by taking the smaller discrepancy from the inter-origin distances distribution. For the value of β (which tunes the interference bias on the death rate in joint model), we chose the one that gave us the smaller area on the efficiency distribution. For the double-stall-aversion model, the fitting procedure is identical and only requires to fix γ.

## Simplified likelihood analysis

We performed a (simplified) likelihood ratio analysis in order to test the better quantitative performance of the combined model. The full likelihood of the models analyzed here is complex, but we have defined 'partial' likelihoods for the joint and the double-stall-aversion model only taking into account the marginal probabilities shown in *Figure 4—figure supplement 1*. Hence, the test evaluates for both models the goodness of the predicted correlation between the efficiency and firing rate of the lost origins and the ones of their neighbors. The likelihood ratio test quantifies how much the prediction of a certain model is better than a reference ('null') model. We chose the double-stall-aversion model as reference (equivalent to setting $\beta = 0$ in the joint model). Specifically, one evaluates

$$L_r = 2 \log \left( \frac{L_{joint}(\gamma,\beta)}{L_{DS}(\gamma,\beta=0)} \right) = 2 \left( l_{joint}(\gamma,\beta) - l_{DS}(\gamma,\beta=0) \right), \tag{8}$$

where $L_X$ are the likelihoods of the two models and $l_X$ are the log-likelihoods. Assuming that $L_r$ is $\chi$-squared distributed (this is generally the case for large samples), we could compute a p-value associated to this test.

## Null birth-death model

We defined a null birth-death model where origin birth-death events in sister species are uncorrelated in order to analyze the divergence of shared origins and compare it with the prediction of the evolutionary model. This model implements birth and death events uniformly, regardless of origin position and firing rate, fixing the number of events for each branch of the simulated phylogenetic tree. These values are taken from the inference reported in *Agier et al., 2018* (shown in Figure 3A of that study and in *Figure 1—figure supplement 1*). The simulation of this model starts with 220 origins (the number of origins inferred for to LA2, the species at the root of the tree). Subsequently, following the structure of the *Lachancea* phylogenetic tree, the simulation proceeds as follows: (i) at each branching point, the genome is copied into two daughters; (ii) for each daughter, the prescribed number of random death and birth events (in this order) is generated on random origins; and (iii) the simulation stops when it reaches the leaves of the *Lachancea* tree.

## Acknowledgements

We are very grateful to Ludovico Calabrese, Simone Pompei, and Orso Maria Romano for the useful discussions. We also thank the editors and reviewers of this manuscript for their constructive and helpful feedback. This work was supported by the Italian Association for Cancer Research, AIRC-IG (REF: 23258).

## Additional information

### Funding

| Funder | Grant reference number | Author |
| --- | --- | --- |
| Associazione Italiana per la Ricerca sul Cancro | IG REF: 23258 | Marco Cosentino Lagomarsino |

The funders had no role in study design, data collection and interpretation, or the decision to submit the work for publication.

## Author contributions
Rossana Droghetti, Formal analysis, Investigation, software, Writing – original draft; Nicolas Agier, data-curation, Formal analysis; Gilles Fischer, Conceptualization, Formal analysis, methodology, Writing – review and editing; Marco Gherardi, Conceptualization, Formal analysis, Investigation, Writing – review and editing; Marco Cosentino Lagomarsino, Conceptualization, Formal analysis, Funding acquisition, Investigation, Supervision, Writing – original draft, Writing – review and editing

## Author ORCIDs
Rossana Droghetti ![ORCID] http://orcid.org/0000-0002-4596-7288
Gilles Fischer ![ORCID] http://orcid.org/0000-0001-5732-2682
Marco Cosentino Lagomarsino ![ORCID] http://orcid.org/0000-0003-0235-0445

## Decision letter and Author response
Decision letter https://doi.org/10.7554/eLife.63542.sa1
Author response https://doi.org/10.7554/eLife.63542.sa2

## Additional files

### Supplementary files
• Supplementary file 1. Results of the simplified log-likelihood tests of the joint and the double-stall-aversion model with the associated p-values. Positive log-likelihood differences favor the joint model (see Materials and methods).

• Transparent reporting form

### Data availability
The code used to run the simulations, together with a readme file, was shared as a Mendeley data repository (https://data.mendeley.com/datasets/vg3r5355bj/2).

The following previously published datasets were used:

| Author(s) | Year | Dataset title | Dataset URL | Database and Identifier |
|---|---|---|---|---|
| Agier N, Stéphane D, Qing Z, Aubin F, Yan J, Van Dijk E, Claude T, Martin W, Cosentino-Lagomarsino M, Fischer G | 2018 | | https://www.ncbi.nlm.nih.gov | SRA - project number: SRP111158. Accessions ranging from SRR5807795 to SRR5807891, SRP111158 |

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

## Appendix 1

### Estimating parameters for the evolution of origin firing rates

This section motivates the model implementation of the evolutionary dynamics of firing rates. In order to quantify the change of origin firing rates over evolutionary times, we studied how the correlation between firing rates of conserved origins behaves as species diverge (*Figure 1—figure supplement 3A*). To quantify the divergence, for each pair of species in the *Lachancea* clade we calculated the Spearman correlation coefficient between the sets of firing rates belonging to corresponding origins in the two species considered (normalized by the species mean firing rate). We found that the more the species are distant, the less these two sets are correlated, which means that origin initiation rates diverge during evolution and origins lose memory of their initial firing rate. The model describes the evolution of firing rates as follows. Every origin changes its firing rate by extracting a new value from the distribution of empirical normalized ones, regardless of their previous firing rate. This process is characterized by a resampling rate $R$, common to all the origins, which defines the probability per unit time that an origin resamples its firing rate. The slope of the correlation coefficient in empirical data defines the speed at which the origin firing rates evolve. Hence, it is possible to fit this specific slope and extract the value of $R$.

In order to do that, we simulated the evolutionary process with unbiased origin death and update of the firing rate. This simulation can be performed without the birth process because the only origins that one needs to consider in computing the Spearman coefficient between two species are the conserved ones. Each simulation started from 225 origins, with firing rates randomly sampled from the empirical set of firing rates, evolving the genomes changing the firing rates with the resampling process described above and removing the origins according to the death rate estimated from the data. By performing several simulations with different values of the extracting rate $R$, it is possible to fit its best value. For each $R$ tested, we ran 1000 simulations for an evolutionary time corresponding to 1.6.

After computing the Spearman correlations between snapshots at different evolutionary times, we performed an exponential fit in order to see which value of the $R$ parameter gave the best agreement with the experimental data, finding the best-fit value $R = 0.92$. *Figure 1—figure supplement 4* shows the trend achieved by the simulation using $R = 0.92$, and it shows a very good agreement between experimental data and simulations.

Note that in *Agier et al., 2018* a similar analysis was carried out in order to verify if the reprogramming of the origins firing rate has an impact on the differentiation of replication timing. The authors analyzed the origin firing time *differences* between conserved replication origins in all pairs of species and found that this difference does not correlate with the phylogenetic distance between species. This finding is apparently in contrast with our results, which suggest that origin reprogramming increases with distance between species. We believe that this discrepancy is due to the higher sensitivity of the Spearman correlation and of the use of species-average normalized firing rates in this study.

### The empirical data falsify the scenario where interference alone drives origin evolution

This section presents a theoretical analysis of the scenario where solely origin interference sets the evolutionary pressure on replication timing profiles. This analysis shows that a description that only takes into account the evolutionary pressure that acts on origin efficiency is not able to reproduce the origins spatial arrangement, a crucial feature in empirical yeast data. To carry out this analysis, we take a 'maximum entropy' approach (see Banavar JR, Maritan A, Volkov I. Applications of the principle of maximum entropy: from physics to ecology. J Phys Condens Matter. 2010;22(6):063101. doi:10.1088/0953-8984/22/6/063101) and infer an effective 'force potential' acting on inter-origin distance by looking at its (assumed equilibrium) distribution. Specifically, the effective potential acting on the origin efficiency starting from the empirical efficiency distribution can be analytically computed from the following formula:

$$H_{\text{eff}}(\text{eff}) = -\log(P(\text{eff})) \tag{S1}$$

where eff is the efficiency, $\text{eff} \in [0, 1]$, and $P(\text{eff})$ the efficiency probability density function.

The above potential, once given the relation between efficiency and distance between origins (*Equation 4*), defines another potential $H_d(d)$ that act on the inter-origin distances. By taking the

exponential of $H_d(d)$ one obtains the expected probability distribution predicted for the distances at equilibrium.

In order to find $H_d(d)$, one must invert *Equation 4* and find $d(\text{eff})$. To accomplish this task, we have approximated the three-body interaction that gives the efficiency with a two-body interaction. This assumption implies that each origin feels the interference of only one of his two neighbors and is effective as long as three-origin interactions can be decomposed in two-origin components. Under this assumption, *Equation 4* becomes

$$d_{i,i+1} = -\frac{v}{\lambda} \log \left[ \frac{\lambda_i + \lambda_{i+1}}{\lambda_i} (e_i - 1) \right] . \tag{S2}$$

Note that origin efficiency (*Equation 4*) also depends on the firing rates of the origin and its neighbor, hence, strictly speaking, one has that

$$H_d(d_{i,i+1}) = H_d(d_{i,i+1}, \lambda_i, \lambda_{i+1}) . \tag{S3}$$

To eliminate the firing rate dependence, we computed an effective potential $H_d'$ on the distance, which averages the effect of the different firing rates. To this end, we used the mean value theorem for integrals as follows:

$$H_d'(d) = \int d\lambda_i d\lambda_{i+1} P(\lambda_i) P(\lambda_{i+1}) H_d(d_{i,i+1}, \lambda_i, \lambda_{i+1}) = H_d(d_{i,i+1}, <\lambda>, <\lambda>) . \tag{S4}$$

In other words, we substituted all the firing rates with the average one $<\lambda> = 1$ since the rates are normalized on the species average. With this simplification, going from $H_{\text{eff}}(\text{eff})$ to $H_d'(d)$ is straightforward and gives

$$d(e) = -\frac{v}{<\lambda>} \log[2(\text{eff} - 1)] , \tag{S5}$$

and

$$H_d'(d) = H_{\text{eff}}(d(\text{eff})) . \tag{S6}$$

From the potential $H_d'$, we can compute the prediction for the equilibrium probability distribution of inter-origin distances

$$P(d) = N \exp(-H_d'(d)) , \tag{S7}$$

where $N$ is a normalization factor. In order to use this calculation on the data, we inferred the expected potential from the efficiency distribution, assuming that the interaction only depends on efficiency, and we then obtained the model prediction for the expected inter-origin distribution based on the efficiency profile. Comparison of this prediction with the empirical inter-origin distance distribution provides a test of the model. This procedure does not require to adjust any model parameter. *Figure 4—figure supplement 2* shows the result of this analysis. The predicted distribution does not match the empirical one. This means that any evolutionary model that assumes a bias based only on the efficiency (in other words, one that takes into account only the evolutionary pressure given by origin interference) cannot reproduce (at steady state) the correct spatial organization of replication origins.

