## [Decision Letter]

**Acceptance summary:**

The reviewers and editors found that the manuscript presents a simple but compelling model that explains the dynamics of replication origin birth and death, which enhances our understanding of the selection pressures that have shaped the distribution of replication origins.

**Decision letter after peer review:**

Thank you for submitting your article "An evolutionary model identifies the main selective pressures for the evolution of genome-replication profiles" for consideration by *eLife*. Your article has been reviewed by 2 peer reviewers, and the evaluation has been overseen by a Reviewing Editor and Naama Barkai as the Senior Editor. The following individual involved in review of your submission has agreed to reveal their identity: Alessandro De Moura (Reviewer #3).

The reviewers have discussed the reviews with one another and the Reviewing Editor has drafted this decision to help you prepare a revised submission.

The reviewers appreciate that the manuscript presents a simple but compelling model that explains the dynamics of replication origin birth and death, which enhances our understanding of the selection pressures that have shaped the distribution of replication origins. However, both reviewers had a series of concerns. Please go through their reviews below. In your revised manuscript please address in particular the major concerns they have raised.

We would like to draw your attention to changes in our policy on revisions we have made in response to COVID-19 (https://elifesciences.org/articles/57162). Specifically, when editors judge that a submitted work as a whole belongs in *eLife* but that some conclusions require a modest amount of additional new data, as they do with your paper, we are asking that the manuscript be revised to either limit claims to those supported by data in hand, or to explicitly state that the relevant conclusions require additional supporting data.

*Reviewer #2:*

The manuscript entitled "An evolutionary model identifies the main selective pressures for the evolution of genome-replication profiles" is an examination of the principles shaping evolution of replication origin placement. Overall I found the manuscript to be engaging and interesting, and the topic of general importance. It is quite compelling that with just two parameters, origin efficiency and distance between origins, a good model can be built to describe the dynamics of origin birth and death. While this work on its own is sufficiently important for publication, it would be very interesting to see whether the model can be updated in the future to address whether there are fork-stalling or origin-generating mechanisms that shape evolution of specific inter-origin spaces. This work provides a very good foundation for such efforts.

I have a few major, general concerns I would like the authors to address.

If I'm interpreting the methods correctly, it seems the parameters used in these simulations, such as mean birth rate, mean death rate, γ, and β, were fit to the data once, and used as point estimates during simulation. If true, I expect the simulations to be yielding estimates of birth and death rates with a much narrower distribution of outcomes than is likely to be realistic given what an appropriate level of confidence in those parameter estimates would be. Could the parameters be fit to data in such a way that we attain an estimate of confidence in the parameter values, from which a distribution could be generated and sampled from during simulation?

Closely related to my prior concern, I would like the authors to demonstrate the general predictive value of their model on out-of-sample data. Can the model be applied to other data on replication timing? Without such an attempt to demonstrate the model's applicability to out-of-sample prediction, the reader cannot ascertain whether the model is overfit to the Lachancea data from Agier et al., 2018. Also, keeps the parameter estimates here from being overfit to better predict origin birth and death events in closely related branches of the Lachancea tree in Figure S1? Are γ and β inferred in a way that accounts for the higher correlation in birth and death events in closer-related branches than in distal branches, or has the fit ignored those correlations?

The authors state that their model identifies selective pressures. The authors imply, and specifically state in lines 238-242, that increased death rate of origins which happen to be nearby highly efficient origins represents selective pressure against the less efficient origins. It isn't until the discussion that the authors raise the possibility that there may simply be a lack of selective pressure to retain inefficient origins that are near highly efficient origins. In my view, it's more likely that selection for the existence of an inefficient origin is simply lower than the drift barrier, so mutagenesis and drift can passively remove such origins over time without the need to invoke selection against inefficient origins.

Figure 3 is intended to show that the stall-aversion and interference model performs better at predicting correlations between efficiency of lost origins and their nearest neighbor. I agree, but I do not think Figure 3 presents a strong case for this conclusion. Figure S6 presents stronger evidence to me. While Figure 3 does qualitatively suggest that the joint model may predict the correlation between neighboring origin efficiency and origin loss better than the double-stall model alone, it almost appears to me that the model with fork stalling and interference has significantly overestimated the correlation. Is there a quantitative way, perhaps using information criteria, though I admittedly am not sure how one would go about doing that with simulations such as these, to demonstrate that the model with both effects has better predictive value than the one with only fork stalling?

There are a couple of assumptions of the model that I would like the authors to examine in further detail. First, that origin birth events occur in the middle of an inter-origin space. I am not aware of evidence pointing to this being a good a priori assumption. Can you re-run the simulations, allowing origins to arise at a random site within the inter-origin space into which it is born? Second, is it reasonable to expect origin firing rates to reshuffle to a new value randomly, without any dependence on their prior rate? Perhaps I'm mistaken, but it seems to me that an origin's firing rate should evolve more gradually, and should have a higher probability of sampling from values near its current value than from values very far from its current value.

*Reviewer #3:*

This paper proposes a novel and relevant evolutionary model that explains many aspects of replication origin statistics in a family of yeast species. It is a step forward in our understanding of the evolutionary pressures that affect the distribution of replication origins in Eukaryotes. I recommend it should be published in *eLife*, provided the authors revise the paper to address the following issue:

1. Many of the conclusions of the paper are based on the claim that the extending the model by adding an efficiency bias to the origin death rate makes the model fit the data better; in particular, they say in line 213 that "the observed huge divergence in efficiency between lost origins and their neighbors is absent in the model simulations."

This is reinforced in line 243, and in other parts of the text. But inspecting Figure 3, the two models (with and without a death rate bias) yield almost identical box-plots; if anything, the box-plots for the lost/nearest fractions of the pure double-stall aversion model seem visually to match the data marginally better. So why do the authors claim that the model with death rate bias is a much better fit? This is far from clear by just inspecting the data. I see no "huge difference" in the plots. There is a difference, but it is far from huge – the differences in the mean are much smaller than the size of the boxes. It seems to me unjustified to use this to choose one model over another. One way to ascertain this is to do rigorous statistical tests to determine if the differences in the means of the simulated and observed data are statistically significant; for example, a t-test.

---

## [Author Response]

Reviewer #2:The manuscript entitled "An evolutionary model identifies the main selective pressures for the evolution of genome-replication profiles" is an examination of the principles shaping evolution of replication origin placement. Overall I found the manuscript to be engaging and interesting, and the topic of general importance. It is quite compelling that with just two parameters, origin efficiency and distance between origins, a good model can be built to describe the dynamics of origin birth and death. While this work on its own is sufficiently important for publication, it would be very interesting to see whether the model can be updated in the future to address whether there are fork-stalling or origin-generating mechanisms that shape evolution of specific inter-origin spaces. This work provides a very good foundation for such efforts.I have a few major, general concerns I would like the authors to address.If I'm interpreting the methods correctly, it seems the parameters used in these simulations, such as mean birth rate, mean death rate, γ, and β, were fit to the data once, and used as point estimates during simulation. If true, I expect the simulations to be yielding estimates of birth and death rates with a much narrower distribution of outcomes than is likely to be realistic given what an appropriate level of confidence in those parameter estimates would be. Could the parameters be fit to data in such a way that we attain an estimate of confidence in the parameter values, from which a distribution could be generated and sampled from during simulation?

We agree that this point is important. To verify consistency of using single values for these parameters, we compared the variability of the local numbers of birth and death events along different branches of the tree obtained from simulations with data. Figure 5—figure supplement 2 shows that simulations and empirical data present similar spreading, especially considering death events. For births, we noticed one relevant outlier for the longest branch of the tree, which could motivate a future deeper analysis. Overall, simulations using constant rates yield a similar distribution of events to data, which appears to be satisfactory, at least to a first approximation. For the bias parameters *γ* and *β*, this additional test is not necessary, as these two parameters were fixed by choosing the values giving the smallest error on these two distributions (hence a constant parameter model, neglecting variations, is sufficiently good for these parameters). This is shown in Figure 3 (panels A and B) of the revised version main text.

Closely related to my prior concern, I would like the authors to demonstrate the general predictive value of their model on out-of-sample data. Can the model be applied to other data on replication timing? Without such an attempt to demonstrate the model's applicability to out-of-sample prediction, the reader cannot ascertain whether the model is overfit to the Lachancea data from Agier et al., 2018.

This is an interesting point but unfortunately no systematic data exist to perform this analysis. Figure 5 of the revised version shows that t-independent predictions apply across the tree. It is important that this model is based on simple global parameters, and not ne tuned on local features of the tree. To emphasize this point, we performed a model t using only a sub-sample of the yeasts, and showed that it leads to the same conclusions. For this analysis we choose the subtree between LADA and LAWA (see Author response image 1) in order to sample a subset of species that were neither too close nor too far compared to the full sample. We anyway expect that the estimated parameters may vary with particular choices of subtrees, also due to sampling effects. This procedure yielded parameter values that are very similar to the t of the full tree. For example the new death (=0.58 *t*^−^1) and the birth (=13.63 *Mbp*^−^1*t*^−^1) rate are really close to the values estimated using the whole phylogenetic tree, and the fitting procedure of the joint model with these new parameters leads to a values for the *γ* and *β* parameters of 2.25 and 1.9 respectively, which, again, are two values really close to the ones obtained with the full tree parameters. We added a comment in the text stating explicitly that we cannot exclude that the values of the birth and death rate, and also the bias parameters *γ* and *β* could be Lachancea-specific, while the conclusions on the relevant evolutionary mechanisms might apply more generally.

**Author response image 1. sa2fig1:** Subtree chosen for the model fit.

Also, keeps the parameter estimates here from being overfit to better predict origin birth and death events in closely related branches of the Lachancea tree in Figure S1? Are γ and β inferred in a way that accounts for the higher correlation in birth and death events in closer-related branches than in distal branches, or has the fit ignored those correlations?

The birth and death rate are inferred as global parameters, ignoring correlations. Despite of this, the model reproduces the higher correlation in birth and death events in closer-related branches than in distant branches as a consequence of the common positions and firing rates of the origins in the ancestor, as Figure 5B of the main text shows. The bias parameters *β* and *γ* are not inferred based on branch data, but on distributions on extant species. We clarified these points in the revised text and the caption of Figure 5.

The authors state that their model identifies selective pressures. The authors imply, and specifically state in lines 238-242, that increased death rate of origins which happen to be nearby highly efficient origins represents selective pressure against the less efficient origins. It isn't until the discussion that the authors raise the possibility that there may simply be a lack of selective pressure to retain inefficient origins that are near highly efficient origins. In my view, it's more likely that selection for the existence of an inefficient origin is simply lower than the drift barrier, so mutagenesis and drift can passively remove such origins over time without the need to invoke selection against inefficient origins.

We fully agree with this point. We have modified the text in order to state early on that the evolutionary bias against weak origins could be simply due to the fact that they are neutral, or that their advantage is not sufficiently high for them to survive drift.

Figure 3 is intended to show that the stall-aversion and interference model performs better at predicting correlations between efficiency of lost origins and their nearest neighbor. I agree, but I do not think Figure 3 presents a strong case for this conclusion. Figure S6 presents stronger evidence to me. While Figure 3 does qualitatively suggest that the joint model may predict the correlation between neighboring origin efficiency and origin loss better than the double-stall model alone, it almost appears to me that the model with fork stalling and interference has significantly overestimated the correlation. Is there a quantitative way, perhaps using information criteria, though I admittedly am not sure how one would go about doing that with simulations such as these, to demonstrate that the model with both effects has better predictive value than the one with only fork stalling?

This point was also raised by reviewer 3. In order to address it, we performed a (simplified) likelihood ratio analysis in order to show the better quantitative performance of the combined model. The full likelihood of our model is complex, but we have de ned partial likelihoods for the joint and the double stall aversion model just taking into account the marginal probabilities shown as box plots in Figure 4 of the revised text (former Figure 3) and in Figure 4—figure supplement 1. In other words, we have tested the predicted correlation between the efficiency and firing rate of the lost origins and the ones of their neighbor for both models. The likelihood ratio test quantifies how much the prediction of a certain model is better than a reference ( null ) model. We chose the double stall aversion model as reference (equivalent to setting *β* = 0 in the full model). See Equation 8, where *L_X_* are the likelihoods of the two models and *l_X_* are the log-likelihoods. Assuming that *L_r_* is *χ*-squared distributed (this is generally the case for large samples), we could compute a P-value associated to this test. Supplementary file 1 shows that the joint model performs better for all the four chosen features. In our view, the qualitative difference shown in Figure 4 may be taken as a stronger argument in favor of the combined model, in the sense that, beyond any quantitative agreement relying on parameters, the additional ingredient is needed to explain the data. We revised the text to highlight this point.

There are a couple of assumptions of the model that I would like the authors to examine in further detail. First, that origin birth events occur in the middle of an inter-origin space. I am not aware of evidence pointing to this being a good a priori assumption. Can you re-run the simulations, allowing origins to arise at a random site within the inter-origin space into which it is born?

We agree that these analyses are useful, and we implemented the analysis of the suggested model variants relaxing specific assumptions. Author response image 2 shows the resulting inter-origin distance distribution for a model where the double stall aversion is implemented, but where newborn origins are placed at uniformly chosen position of the associated interval. The figure shows that relaxing the assumption that new origins are born at mid points has consequences on the distance distribution, irregardless of the value of the parameter *γ*. We interpret this as the result of a faster (hence undetectable in our data) evolutionary process that counter selects origins far from midpoints. The reason for placing newborn origins at midpoints can then be justified by the results of Newman et al.: in order to minimize the global double stall probability the origins perform several attempts within a tree branch, of which we observe only the successful ones. This choice is also justified in the data by Figure 1—figure supplement 2, which shows that birth events show a strong bias for being close to mid points of their associated intervals. We added this figure as a supplementary figure to justify our assumption.

**Author response image 2. sa2fig2:** The "uniform draw" model does not reproduce the inter origin distance distribution. The figure compares the empirical distance distribution (red point line) with the one resulting from the simulation of the uniform draw model (blue bars) with γ=10. We choose this value because for γ=10 the error made on this distribution has already saturated to the lower reached level. The model tested here cannot reproduce the inter origin distance distribution.

Second, is it reasonable to expect origin firing rates to reshuffle to a new value randomly, without any dependence on their prior rate? Perhaps I'm mistaken, but it seems to me that an origin's firing rate should evolve more gradually, and should have a higher probability of sampling from values near its current value than from values very far from its current value.

We originally started this project with the same idea, modeling rate evolution as a diffusion process. However, on the evolutionary time scales associated to the Lachancea data, it turns out that this expected gradual evolution is not visible, leaving the model with too many extra parameters (a firing rate diffusion constant, and bounds to set the empirical distributions) that are very di cult to estimate. Indeed, Author response image 3 shows the firing-rate distributions of the conserved (thus older) origins and of newborn (younger) ones. The two distribution are similar (within their errors), suggesting that (on the observation time scale) newborn origins already take a firing rate sampled from the steady-state distribution. This is condition that is not generally met under a simple diffusive process. This is also shown in Figure 1—figure supplement 3 of the revised text, which shows that the cumulative probability of the firing rates of newborn and conserved origins are similar across the different Lachancea species. In light of these results, we reverted to a simpler parameter-poor version of the model. Possibly this choice can be improved upon in the future, when additional measurements will be available. We addressed this point in the revised text.

**Author response image 3. sa2fig3:** Newborn origins and conserved ones have similar distributions of firing rates. The plot shows the probability distribution functions for the normalized firing rates for all the origins in the ten Lachancea species (red diamonds) and for those origins which have been gained in the terminal branches belonging to the six sister species (green triangles). The two distributions are similar, supporting the idea that the firing rates of the new gained origins are at steady state.

Reviewer #3:This paper proposes a novel and relevant evolutionary model that explains many aspects of replication origin statistics in a family of yeast species. It is a step forward in our understanding of the evolutionary pressures that affect the distribution of replication origins in Eukaryotes. I recommend it should be published in eLife, provided the authors revise the paper to address the following issue:1. Many of the conclusions of the paper are based on the claim that the extending the model by adding an efficiency bias to the origin death rate makes the model fit the data better; in particular, they say in line 213 that "the observed huge divergence in efficiency between lost origins and their neighbors is absent in the model simulations."This is reinforced in line 243, and in other parts of the text. But inspecting Figure 3, the two models (with and without a death rate bias) yield almost identical box-plots; if anything, the box-plots for the lost/nearest fractions of the pure double-stall aversion model seem visually to match the data marginally better. So why do the authors claim that the model with death rate bias is a much better fit? This is far from clear by just inspecting the data. I see no "huge difference" in the plots. There is a difference, but it is far from huge – the differences in the mean are much smaller than the size of the boxes. It seems to me unjustified to use this to choose one model over another. One way to ascertain this is to do rigorous statistical tests to determine if the differences in the means of the simulated and observed data are statistically significant; for example, a t-test.

This point was also raised by reviewer 2. For the description and results of our analysis please refer to the previous section and to Supplementary file 1.